# Droplet Microfluidic Optimisation Using Micropipette Characterisation of Bio-Instructive Polymeric Surfactants

**DOI:** 10.3390/molecules26113302

**Published:** 2021-05-31

**Authors:** Charlotte A. Henshaw, Adam A. Dundas, Valentina Cuzzucoli Crucitti, Morgan R. Alexander, Ricky Wildman, Felicity R. A. J. Rose, Derek J. Irvine, Philip M. Williams

**Affiliations:** 1Molecular Therapeutics and Formulation, School of Pharmacy, University of Nottingham, Nottingham NG7 2RD, UK; charlotte.henshaw@nottingham.ac.uk; 2Advanced Materials and Healthcare Technologies, School of Pharmacy, University of Nottingham, Nottingham NG7 2RD, UK; Adam.Dundas1@nottingham.ac.uk (A.A.D.); Morgan.Alexander@Nottingham.ac.uk (M.R.A.); 3Centre for Additive Manufacturing, Department for Chemical and Environmental Engineering, Faculty of Engineering, University of Nottingham, Nottingham NG7 2RD, UK; valentina.cuzzucolicrucitti1@nottingham.ac.uk (V.C.C.); Ricky.Wildman@nottingham.ac.uk (R.W.); 4Regenerative Medicine and Cellular Therapies, School of Pharmacy, University of Nottingham, Nottingham NG7 2RD, UK; Felicity.Rose@Nottingham.ac.uk

**Keywords:** microparticle, microfluidics, particle synthesis, biomaterials, biodegradable, interfacial tension, micropipette, surfactants

## Abstract

Droplet microfluidics can produce highly tailored microparticles whilst retaining monodispersity. However, these systems often require lengthy optimisation, commonly based on a trial-and-error approach, particularly when using bio-instructive, polymeric surfactants. Here, micropipette manipulation methods were used to optimise the concentration of bespoke polymeric surfactants to produce biodegradable (poly(d,l-lactic acid) (PDLLA)) microparticles with unique, bio-instructive surface chemistries. The effect of these three-dimensional surfactants on the interfacial tension of the system was analysed. It was determined that to provide adequate stabilisation, a low level (0.1% (*w*/*v*)) of poly(vinyl acetate-co-alcohol) (PVA) was required. Optimisation of the PVA concentration was informed by micropipette manipulation. As a result, successful, monodisperse particles were produced that maintained the desired bio-instructive surface chemistry.

## 1. Introduction

Microparticles have been a matter of interest across many fields for approximately 50 years, due to the diversity offered in terms of material, morphology, and shape combinations, allowing them to be tailored to the application at hand [1,2]. Most notably, microparticles are of significant interest in the fields of drug delivery and tissue engineering. Microparticles loaded with drugs, growth factors or proteins, are able to achieve specific and controlled delivery to cell cultures or as treatments, with the release being tuned further by adjusting the size and internal structure of the particles [2,3,4,5,6,7,8,9,10,11]. Alternatively, they can be used as a tool to direct stem cell fate, modulate the phenotype of other cells or to provide a realistic 3D environment for culture by forming scaffolds [3,12,13,14]. Their abilities in this role can be further enhanced by introducing specific chemistries, topographies or porosity [12,13,15].

For these applications, the chosen materials must be biocompatible and for many, biodegradability is a must. As a result, poly(lactic acid) (PLA) and poly(lactic acid-*co*-glycolic acid) (PLGA) are two of the most common choices for microparticles, since their biocompatibility and biodegradability have been long understood [16]. Additionally, they are compatible with a large range of additives, both commercial and bespoke, making them an ideal base for microparticle design. However, as systems become more complex, greater levels of optimisation are required. 

Particles are commonly produced through solvent evaporation emulsification. While this method provides a quick and simple way to obtain particles, the particles are highly polydisperse [17,18] and will often need filtering before use. Furthermore, as systems become more complex, it may require multiple stages to obtain the desired particles [17,19]. Generation through droplet microfluidics overcomes these difficulties, producing highly monodisperse particles in a scalable and reproducible manner. Even highly complex particles can be produced in one-step methods [1,6,20,21]. Flow-focusing geometries can offer the most homogenous particles and are acknowledged as the best option for scaled-up production. However, non-trivial optimisation steps are required to obtain the desired particles [20].

In a microfluidic system the interfacial and viscous forces dominate over the inertial forces during the emulsion formation process [21,22,23]. Controlling these is critical to forming and maintaining emulsions. Even within stable systems, small changes in the balance of these forces can dramatically alter the size of the resulting drops [21,24,25]. Adjusting the interfacial tension to improve stability can be achieved by the addition of surfactants. It is not always possible, however, to know the effect of the addition of a surfactant without studying it directly in the system, particularly when the interfacially-active species are complex three-dimensional polymers.

Methods to measure both static and dynamic interfacial tensions using micropipette manipulation were developed 20 years ago [26]. Since then, it has been employed to measure water–air and water–oil interfaces with the addition of alcohols, salts and other surface-active materials, including assessing lung surfactants [26,27,28,29,30]. Micropipette interfacial tension measurements rely on the fact that an interface formed within a taper will have a single equilibrium position, and so a defined radius of curvature, *R_c_*, for any applied pressure, *ΔP*. The Young–Laplace equation then allows calculation of interfacial tension, *γ* [26,27].

Micropipette manipulation can be an extremely valuable tool in understanding interfaces in microfluidic systems for several reasons. Firstly, only small quantities of material are required (picolitres per test), allowing detailed testing of new, bespoke materials which are often only available in very small research scale quantities. Secondly, interfacial tension values can change with the scale at which they are measured [31,32]: by using micropipette methods the interfaces are of the same order of magnitude as in the microfluidics system. Finally, the system mirrors that used in the microfluidic production, as can be seen in Figure 1. Other methods may be unable to include all components, for example solvents may have to be replaced/changed if those in the microfluidic system are too volatile. This allows the contribution from each component of the emulsification system to be investigated, and their interactions accounted for. Therefore, it is more suitable for optimisation than traditional methods such as pendent drop or Wilhelmy Plate. 

Additionally, the formation of single droplets to study their transition to solid particles can also be carried out using the micropipette [27,33,34,35]. Drops are observed as the solvent diffuses out of the droplet and enters the surrounding water phase, allowing estimation of the time needed for complete solvent removal, and so to prevent aggregation [33,34,36]. Together these studies avoid the need for lengthy trial-and-error approaches to achieve stable microfluidic production. Micropipette experiments have been used to inform microfluidic processing previously [37], but not to the extent shown in this work.

Poly(vinyl alcohol) (PVA) is frequently used to stabilise PLA/PLGA particles, due to the benefits of its polymeric structure [5,12,13,14,15,18,38]. However, in instances where the particle surface chemistry is important for the application, PVA will dominate this particle characteristic and so screen its effects. Furthermore, its polymeric nature means that it is difficult to remove from the surface; so residual surfactant will often be present even after multiple washes [39]. Other commercially available, larger molecular weight surfactants also present similar issues [40]. This can be overcome by producing bespoke surfactants which exhibit the desired chemistry. 

High-throughput screening (HTS) libraries have been employed to identify new bio-instructive polymer materials targeted for use in medical-based applications. These include materials which can: resist bacterial attachment [39,41,42,43], resist fungal colonisation [44], alter cell phenotype, direct stem cell fate or maintain its pluripotency [45,46,47,48,49]. These libraries comprise 1000s of unique (meth)acrylate copolymers, allowing suitable biomaterials to be designed and synthesised with a frequency that is not possible through other methods. Recently, these materials have been formed into microparticles in order to move from the 2D biological-based assessment of these HTS platforms, which confirm the materials’ predicted activity at larger scales, to 3D testing. In initial studies, monomer droplets were formed using droplet microfluidics and polymerised before collection [39], with PVA used to stabilise drops. However, to overcome the undesirable effects of this surfactant, “hit” chemistries from the libraries were later formed into bespoke surfactants with comb-graft structures. The new surfmers (polymer surfactants) were able to not only produce and maintain the stability of droplets during production, but also provide a complete unimolecular covering of the “hit” material across the surface of the particles such that the biological response was maintained in 3D [50].

In this paper, a new strategy has been proposed for the optimisation and development of biodegradable microparticles functionalised with designed polymer surfmers. Two example “hit” chemistries exhibiting specific, target bio-instructive characteristics were identified using HTS methods, these were ethylene glycol phenyl ether acrylate (EGPEA) and tetrahydrofurfuryl acrylate (THFuA). These have been synthesised into bespoke surfmers by co-polymerising with poly(ethylene glycol) methyl ether methacrylate (mPEGMA) in a ratio of approximately 90:10 hit material to mPEMGA to give: EGPEA-co-mPEGMA and THFuA-co-mPEGMA (Figure 1). We propose that a two-surfactant system can be used to modulate the interfacial properties of an oil-in-water microfluidic system in order to generate stable emulsion formation, and in turn monodisperse particles, whilst retaining important chemical surface moieties that have been shown to modulate cell behaviour. To test this we used micropipette characterisation to observe the changes in interfacial tension that emerge when adding increasing amounts of PVA and surfmer. We examine the effectiveness of the surfmers relative to the established surfactant, and discuss how combining them may be advantageous, particularly when the surface of the particle is important biologically. The resulting particles were characterised using SEM and time-of-flight secondary ion mass spectrometry (ToF-SIMS) to evaluate the success of this approach in developing unique biodegradable particles with cell-instructive surfaces. 

## 2. Results

Our target applications rely on the particles’ ability to degrade, e.g., enabling delivery of a payload. The target core material chosen in this study was poly(d,l-lactic acid) (PDLLA). The strategy adopted was to create the dispersed phase by dissolving pre-synthesised PDLLA and one of the two bespoke surfmers (i.e., EGPEA-co-mPEGMA or THFuA-co-mPEGMA) in ethyl acetate, paired with a continuous phase of distilled water, with or without PVA. 

As interfacial tension is one of the main driving forces for successful droplet production in microfluidics, it was vital to assess how the bespoke surfmers influenced this property. Static equilibrium interfacial tension measurements were made using the micropipette, where a set PDLLA concentration (5% (*w*/*v*)) in ethyl acetate was used against milliQ water, whilst the chosen surfactants were added to the appropriate phase. The use of the core polymer was important to understand the surfactant effects on the whole system. Equally, while the general effects of PVA on interfacial tensions are well known, it was necessary to quantitatively understand how it interacted with the other components, especially the surfmers. Through these studies, the optimum concentration of each surfactant was determined. The effect on interfacial tension of the novel surfmers on the water interface with 5% (*w*/*v*) PDLLA in ethyl acetate is shown in Figure 2A. 

EGPEA-co-mPEGMA did not significantly reduce the interfacial tension, rather, it fluctuated around the value exhibited by the surfactant-free system. In contrast, THFuA-co-mPEGMA produced a slight, but gradual, decrease in the interfacial tension from a value of 6.00 ± 0.80 mN m^−1^ (no surfactant) to 4.89 ± 0.04 mN m^−1^ (2% surfactant). While this decrease in tension would improve stability in the microfluidics system, the concentration needed was so high in relation to the core polymer concentration (5% (*w*/*v*) PDLLA) it would represent a significant proportion of the final particle. Thus, the surface covering is likely to be greater than a unimolecular coating. Furthermore, as this material is non-degradable a significant coating may restrict the decomposition of the final particles.

The ideal composition to obtain the required thickness of surface coating is determined by the final particle size; in this case, necessitating between 0.1% and 0.2% (*w*/*v*) concentration of the surfmers. However, using surfmer concentrations in this range produced insufficient stabilisation, as shown by flow diagrams in Appendix A. 

In order to obtain the necessary droplet stability, PVA was then added into the water phase. The data in Figure 2B show the effect of adding PVA to the continuous phase with both no surfmer and with fixed concentrations (0.1% (*w*/*v*)) of individual surfmers in the dispersed phase. PVA delivered a significant reduction in interfacial tension as the concentration increased from 0% to 2%, both with and without the presence of a bespoke surfmer. Common concentrations of PVA in the water phase used in microparticle formation are between 0.5% and 2% (*w*/*v*). From the trend in Figure 2B, the optimum concentration to minimise the interfacial tension was between 1% and 1.5% (*w*/*v*) PVA. However, to preserve the desired surface chemistry, the PVA concentration needed to be minimised. Thus, a weak (0.1% *w*/*v*) PVA solution was used as the continuous phase to improve the stability of emulsions in a high-throughput droplet microfluidics setup. 

From Figure 2B, combining THFuA-co-mPEGMA with PVA, in the range where PVA ≤ 0.1% (*w*/*v*) provided a greater reduction in interfacial tension than either EGPEA-co-mPEGMA was used in combination with PVA ≤ 0.1% (*w*/*v*) or when PVA was adopted as a single-surfactant system. This observation was confirmed by applying a two-way ANOVA analysis to the data. The difference here is small, and for EGPEA-co-mPEGMA there is no significant change. However, the surfmers are required for their key role in providing the surface coverage of the bio-instructive chemistry. 

Particles were collected in a collection vessel containing DI water and then centrifuged to separate from the water and then analysed using SEM, as seen in Figure 3, where particles produced with PVA alone are shown for comparison. The sizes, as measured by SEM, of the particles produced with the different surfmers were all similar, and the particles were all monodisperse. These results confirm that the addition of 0.1% (*w*/*v*) PVA to the continuous phase is sufficient to produce stable particles if used alongside the bespoke surfmer in the dispersed phase.

Particles were shown to be slightly agglomerated, which is due to the collection method. In order to prevent aggregation of particles during collection, sufficient time must be allowed for the solvent to be removed. Single-particle studies were conducted using the micropipette to record the time taken for solvent removal to be completed from different sized droplets, defined as the particles reaching a constant size. This was performed for drops with a 30 to 80 µm initial radius and for PDLLA concentrations of 1%, 5% and 10% (*w*/*v*) in ethyl acetate. Representative times for the drops to achieve constant size are shown in Figure 4 as individual points and are compared to the theoretical time calculated. 

The theoretical estimates for removal times, depicted by the solid lines Figure 4, were calculated using the Epstein–Plesset model for droplet dissolution from the averaged diffusion rate calculated from several drops. The overall agreement between the measured and calculated values is good and shows that this model provides a reliable method of estimating the necessary collection time that needs to be applied to different systems to prevent aggregation. Using the observed radius from the microfluidic experiments, where the drops at the point of generation were observed to be approximately 40 µm in radius, the expected time needed for solvent removal in the collection vessel was predicted to be approximately 6 s.

To ensure that particles produced retained the surface functional moieties from the polymer surfmers, ToF-SIMS data were acquired to locate unique ions on the surface of the particles using both spectra and chemical mapping analysis (Figure 5). 

Unique ions were identified using ToF-SIMS, allowing the different surfactant structures used in the microfluidic particle preparation to be confirmed. C_6_H_5_^+^ and C_5_H_9_O^+^ were identified as unique ions for EGPEA-co-mPEGMA and THFuA-co-mPEGMA, respectively. C_4_H_9_O^+^ was shown to represent PVA. PVA could be seen to be present in both particles with THFuA-co-mPEGMA and EGPEA-co-mPEGMA at low levels, 4% and 10%, respectively, demonstrating that the surface functionality is dominated by the target chemistry introduced into the dispersed phase.

## 3. Discussion

As the concentration of THFuA-co-mPEGMA increases, the interfacial tension is reduced slightly, while no significant change is seen for EGPEA-co-mPEGMA. This suggests that these are mild surfactants. This conclusion was supported by the results of a previous study using similar comb-graft surfmers in which (meth)acrylate based, hydrophobic, and bio-instructive “hit” monomers from the HTS survey were co-polymerised with mPEGMA; the molar ratio if the latter was present in the final co-polymer in the range ~5–15%. To successfully generate emulsions by applying these surfmers required concentrations approaching 2% (*w*/*v*), when used in a solvent-free droplet microfluidics process [50]. However, in these cases, the dispersed phase was comprised of only hexanediol diacrylate monomer, which formed the core of the particle, the surfmer and a photo-initiator, so the lack of solvent meant that the ratio of surfmer to core stayed constant throughout the whole process. In this study, the dispersed phase contained the pre-synthesised PDLLA and ethyl acetate. As the solvent was removed, the PDLLA and surfactant concentrations increased to a final composition of approximately 98% and 2%, respectively. Had an initial surfmer concentration of 2% (*w*/*v*) been used, as in the comparative study, with 5% PDLLA and solvent, then the final particle composition would be approximately 30% surfmer. To maintain biodegradability of the particles, the target of the surfmers is to provide a uni-molecular coverage on the particles, therefore necessitating the reduced of surfmer concentration to an initial value of 0.1% (*w*/*v*).

Comparing single-surfactant systems (Figure 2A and PVA in Figure 2B), it is clear PVA is more effective in the high concentration range than the surfmers are, whilst in the low concentration range (<0.5% (*w*/*v*)), they behave very similarly. When small volumes of surfmer were used with high PVA volumes, the interfacial tension values matched those of PVA on its own (Figure 2B). This is to be expected since the concentration of PVA far exceeds that of the surfmer. However, for low PVA concentrations, the combination of PVA and surfmer showed a greater reduction in interfacial tension than either did alone, specifically 0.1% (*w*/*v*) THFuA-co-mPEGMA with 0.1% (*w*/*v*) PVA, which produced the lowest interfacial tension value within the concentration limits. At these low concentrations it cannot be said whether the surfmer or PVA dominate. Generally, the comparison of PVA and the surfmers shows that the surfmers are clearly much weaker surfactants than PVA. However, they are amphiphilic enough to achieve the desired coating on the particle surfaces, as demonstrated in Figure 4, thus satisfying their purpose.

The two surfmers show very different behaviour in terms of spread of the results: in most cases the standard deviation on the THFuA-co-mPEGMA measurements is an order of magnitude lower than EGPEA-co-mPEGMA. The difference likely arises from “pinning” of the interface, which refers to the interface appearing to stick to the walls of the pipette rather than moving smoothly when the pressure changes. This results in a slight deformation of the interface or a small deviation from the expected position. This affects individual radius of curvature, *R_c_*, measurements which in turn impact the individual and therefore the overall interfacial tension measurement. The effects of pinning are increased for materials with a strong affinity for glass and high viscosity, but these have been mitigated in this study by allowing systems to fully reach equilibrium prior to measurements being taken and measurement of the interfaces curvature being carried out away from the pipette walls. Both THFuA-co-mPEGMA and EGPEA-co-mPEGMA have the same amphiphilic component, which could suggest the difference in behavior is a result of the bio-instructive chemistry component. 

The produced particles are monodispersed, confirmed by the coefficient of variance being a maximum of 5.7%, which was for those produced with only PVA. These were also the smallest, 19.9 ± 1.1 µm compared to 23.1 ± 1.2 µm and 23.7 ± 1.3 µm for particles with THFuA-co-mPEMGA and EGPEA-co-mPEGMA, respectively. The PVA-only particles are used as a comparative system. Attempts were made to produce particles in the microfluidic apparatus using pure PDLLA for comparison to the surfactant included analogues. However, it proved impossible to generate stable droplets so this comparison could not be made.

Despite using the same concentration of PVA and surfmer in their respective phases, the surfaces of the resulting particles are dominated by the surfmer chemistry (Figure 5). This may be due to surfmers being in the dispersed phase and so concentrating on the initial droplet interface or that PVA, by being in the continuous phase, is diluted and so, even though they all have a similar driving force the result is a lower coverage of PVA on the final particle. An alternative explanation is that PVA contributes to the initial stability, as evidenced by unsuccessful emulsion production in its absence (Appendix A), and the surfmers migrate to the surface over longer timescales to provide lasting stabilisation, with their structure providing steric stability to prevent aggregation during the later stages of production. The competing effects could be investigated through their dynamic interfacial tensions. 

Though the performance of PVA at the chosen concentration is relatively weak, it satisfied the needs of the system—as shown by sustained, monodisperse particle production. PVA was chosen as it is a readily available and well-understood surfactant, presenting a good starting point, and demonstrated the need for a secondary surfactant. While only present at very low levels on the particles’ surfaces, it is acknowledged that this is still not ideal. As such the aim would be to use these findings to move to a bespoke replacement for PVA. Since it has been shown that mild surfactant behaviour in the continuous phase is sufficient, it may be possible to achieve stability by adapting these surfmers to be water-soluble. The adaptation of the surfmers also provides the opportunity to ensure all components are biodegradable.

Viscosity was not explicitly investigated in this work as when considering the relations used to describe microfluidics systems, for example the capillary number, interfacial tension and viscosity both scale linearly (Ca ∝ μγ, where *µ* is viscosity) but the impact from the addition of surfactants on viscosity, compared to that on the interfacial tension, is negligible [21,22,23,24]. Since achieving a surface coverage of the bio-instructive chemistry remained a key aim, understanding and directing the use of the surfmers was the priority. The viscosity change in the dispersed phase, due to the addition of the surfmer, can be considered negligible compared to the contributions of the core polymer and solvent, whose proportions were initially selected to satisfy this limit. However, it may be that that the improved stability observed was due to the viscosity ratio of the two phases changing due to the addition of PVA producing a small change in the continuous phase viscosity. 

While the micropipette interfacial tension studies were able to match the conditions in the microfluidics, for the solvent removal studies, there are factors that cannot be accounted for. These include the effects of flow and saturation in the microfluidics system. Micropipette studies are carried out in sink conditions, which may allow faster solvent removal than in the chip or tubing of the microfluidics system. Conversely, the effects of flow in microfluidics could speed up removal. To an extent, these effects will counter one another. However, as a precaution, advised times should start when drops leave the tubing and entre the collection vessel where conditions are closer to those in the micropipette. The presence of aggregation, should the particles be collected too quicky, suggests there is residual solvent still being removed. Alternatively, it may be that solvent present in the continuous phase is sufficient to dissolve the outer of the particle enough to encourage sticking. In either case this presents an opportunity for further improvements to be investigated. 

The results shown here demonstrate the validity of micropipette techniques for optimising droplet microfluidics processes, with both stability and solidification considered. These methods can be applied to assess oil-in-water and water-in-oil systems with a diverse range of surfactants in either phase. As such, optimisation in this manner could be utilised across the full breadth of emulsion and particle applications where droplet microfluidics can be applied. The surfmers produced provide a reliable way to obtain functionalised particles in a single step. By synthesising surfmers with an appropriate “hit” chemistry the particles’ surface chemistry can be tailored for various applications, such as directing cell behaviour which is of particular use in bio-instructive scaffolds.

In addition to replacing PVA, future work with this system will follow two paths: One is to perform biological assessment with these particles to confirm the surface chemistry, clearly displayed in the ToF-SIMS results, remains able to influence biological function in this format. The second route is to continue to develop biodegradable particles applicable to diverse therapies. Within this both chemistry and morphology will be considered. 

## 4. Materials and Methods

### 4.1. Materials

Throughout the experiments, poly(d,l-lactic acid) (PDLLA), Mn 47,000 gmol^−1^, from Evonik Rohm GmbH (Darmstadt, Germany), poly(vinyl acetate-co-alcohol) (PVA) M_w_ 25 kDa, 88% hydrolysed from Sigma-Aldrich (St. Louis, MO, USA,)and ethyl acetate from Fisher Scientific (Waltham, MA, USA) were used, without modification. 

Surfmers were produced in house using poly(ethylene glycol) methyl ether methacrylate (mPEGMA), Mn = 300, ethylene glycol phenyl ether acrylate (EGPEA), tetrahydrofurfuryl acrylate (THFuA), and 2,2′-azobis (2-methylpropionitrile) (AIBN, 98%). These materials were purchased from Sigma Aldrich and used without further purifications. The catalytic CTA, bis[(difluoroboryl)diphenylglyoximato] cobalt (II) (PhCoBF) (DuPont, DE, USA). Cyclohexanone and heptane used as solvents for the synthesis and precipitations, respectively, were used as received and supplied by Fisher Scientific.

### 4.2. Synthesis and Characterisation of the Surfactants

Both EGPEA-co-mPEGMA and THFUA-co-mPEGMA copolymers were synthesised using catalytic chain transfer polymerisation (CCTP) with the following protocol. The appropriate quantities of the monomers required to reach the targeted molar ratios (90:10 mol/mol) were dissolved in cyclohexanone in a 1:3 *v*/*v* ratio. The initiator and chain transfer agent were added in the reaction vessel with monomers and solvent. A PhCoBF concentration in cyclohexanone of 850 ppm (0.89 mg/mL) was used. AIBN (0.5% *w*/*w*, with respect to the total monomer mixture used) was dissolved in cyclohexanone and degassed separately, prior to being added to the reaction mixture. Finally, the reaction vessel and the AIBN solution were degassed via purging with argon, using a standard Schlenk line technique for at least 1 h in an ice bath, then AIBN was introduced to the reaction mixture. The reaction was carried out at 75 °C for 18 h with agitation. Polymer purification was conducted via precipitation of the crude reaction solution into an excess of heptane with a non-solvent: reaction media ratio of 5:1 *v*/*v* in order to enhance the precipitation process. Finally, the precipitated materials were collected and left in a vacuum oven at 25 °C for at least 24 h.

^1^H-NMR spectroscopic analysis was performed on the crude polymerisation solution to determine polymer conversion and on the precipitate to establish the monomer ratio of the final copolymer composition. ^13^C-NMR was performed on the precipitate to confirm the chemical composition of the materials. ^1^H NMR and ^13^C-NMR spectra were recorded at 25 °C using a Bruker DPX−300 spectrometer (400 MHz) (Bruker, Billerica, MA, USA). Chemical shifts were recorded in ppm. Samples were dissolved in deuterated chloroform (CDCl_3_) to which chemical shifts are referenced (residual chloroform at 7.26 ppm and 77 ppm). To evaluate the molecular weight and molecular dispersity of the materials, the purified samples were dissolved in HPLC-grade THF for GPC analysis. GPC analysis was performed by using an Agilent 1260 Infinity instrument (Agilent, Cheadle, UK) equipped with a double detector with the light scattering configuration. 2 mixed C columns at 25 °C were employed, using THF as the mobile phase with a flow rate of 1 mL/min. GPC samples were prepared in HPLC-grade THF and filtered previous to injection. Analysis was carried out using Astra software (Wyatt Technology, Goleta, CA, USA). The number average molecular weight (M_n_) and polydispersity (Ð) were calculated using PMMA for the calibration curve. 

THFuA-co-mPEGMA. ^1^H-NMR: 3.90–3.66 (^3^H, OC*H*_2_C*H*O, m), 3.63–3.45 (^4^H, CHOC*H*_2_ and C=OOC*H*_2_ (mPEGMA), m), 3.43 (^18^H, C=OOC*H*_2_CH_2_O and (OCH_2_CH_2_O)_4_, m), 3.14 (^3^H, OC*H*_3_, s), 1.94–1.72 (^3^H, OC*H*_2_CH_2_CH*H*, m), 1.51 (^1^H, OC*H*_2_CH_2_C*H*H, m). ^13^C-NMR: 174 (C=O), 76.22 (*C*HOC_3_H_6_), 71.74 (*C*H_2_OCH_3_), 70.48 ((O*C*H_2_*C*H_2_)_4–5_), 68.54 (O*C*H_2_CH_2_, O*C*H_2_CH), 68.07 (OCH_2_*C*H_2_), 59.08 (OCH_3_), 41.95 (CHO*C*H_2_C_2_H_4_), 28.18 (CHOCH_2_*C*H_2_CH_2_), 25.64 (CHOCH_2_CH_2_*C*H_2_). GPC: 18 900 g/mol, Ð 2.22.

EGPEA-co-mPEGMA. ^1^H-NMR: 7.22 and 6.87 (^5^H, C_5_*H*_5_, m), 4.30 (^4^H, C=OOC*H*_2_*,* m) 4.04 (^2^H, OCH_2_C*H*_2_, m), 3.60 (^18^H, C*H*_2_C*H*_2_O and (OC*H*_2_C*H*_2_)_4_O, m), 3.40 (^3^H, OCH_3_, m). ^13^C-NMR: 174 (C=O), 129, 121.13 and 114.46 (C_6_H_5_), 82.56 (O*C*HC_9_H_12_), 71.74 (*C*H_2_OCH_3_), 70.48 ((O*C*H_2_*C*H_2_)_4–5_), 68.54 (O*C*H_2_CH_2_), 68.07 (OCH_2_*C*H_2_), 59.08 (OCH_3_). GPC: 17,000 g/mol, Ð 2.28.

### 4.3. Static Equilibrium Interfacial Tension Measurements

Briefly, the micropipette manipulation setup comprised a 1 mm × 2 mm × 20 mm open-ended glass chamber mounted over an inverted microscope (Axiovert 100, Zeiss, Berlin, Germany), with a camera (Allied Vision Technologies, Pike, QC, Canada) linked to a LABVIEW program capable of recording images and videos. Glass pipettes were formed from borosilicate glass capillaries from A-M Systems by first pulling, and then forging a clean tip using P−97 Micropipette Puller (Sutter Instruments, Novato, CA, USA) and a MF−830 microforge (Narishige, Tokyo, Japan), respectively. 

The chamber was filled with the continuous phase, here milliQ water with or without PVA. The pipette was front filled, first with a solvent saturated plug, in this case water saturated with ethyl acetate, to prevent evaporation; then front filled with the dispersed phase, 5% (*w*/*v*) PDLLA in ethyl acetate with varied concentrations of the bespoke surfactants. The pipette was connected via tubing to a syringe and pressure sensor (Validyne CD223) to control and record the applied pressure. It was then mounted inside the chamber where an interface formed between the dispersed and continuous phases replicating the microfluidics system (Figure 1).

Interfacial tension was measured first with no surfactant to obtain a starting value (5% (*w*/*v*) PDLLA in ethyl acetate with milliQ water). Surfactant concentration in the dispersed phase was increased from 0% to 2% (*w*/*v*) with respect to ethyl acetate, for each of the two bespoke surfmers. PVA was added to the continuous phase in concentrations up to 2% (*w*/*v*) and measured in conjunction with a fixed surfmer concentration (0.1% (*w*/*v*)) in the dispersed phase. 

A fresh interface was formed, as detailed previously, [26,27] before incrementally increasing the pressure applied, and the interface being allowed to reach its equilibrium position. An image was acquired before the pressure was increased again, and the process repeated. Once the interface approached the tip of the pipette the pressure was decreased, and images taken as the interface receded. A minimum of 20 images were taken for each repeat, with three repeats made for each measurement.

Images were analysed using Fiji ImageJ, applying the manual chord method to determine the radius of curvature for each image [26]. The measurements and pressure extracted from the image were read into a MATLAB script to produce a plot of pressure as a function of reciprocal radius (*P* vs. 1/*R_c_*) and the Curvefit tool applied to extract the gradient, and thus the interfacial tension value. 

### 4.4. Solvent Removal Studies

The instrument setup for this was similar to that used for the interfacial tension measurements except two narrow, almost parallel pipettes replaced the single tapered pipette, and were placed opposite one another in the chamber. The first was filled with the dispersed phase, as described for the interfacial tension measurements, the second filled with the continuous phase on entry to the chamber. A droplet was extruded from the first pipette and transferred to the second, as per the single microdroplet catching method, where it was held stationary as the solvent diffused out of the droplet. A video of the droplet was recorded until no change in drop size was seen [27]. 

For each droplet, the video was read into a MATLAB script, which processed individual frames, identifying and measuring the drop radius by utilising the Hough Transform. For each processed frame, the time and radius were plotted. An example size profile of a drop is given in Appendix A. In each case, the profile was used to extract the rate of dissolution and from this, the diffusion coefficient of the dispersed-phase solvent (ethyl acetate) into the continuous (aqueous) phase determined. Since the diffusion coefficient was found to be approximately constant across the polymer concentrations and drop sizes, an average value was taken. The Epstein–Plesset model was applied [36,37,51], using the averaged diffusion coefficient, to calculate the theoretical estimate of the time taken for solvent to be removed from drops of increasing radii from 1 to 100 µm for 1%, 5% and 10% (*w*/*v*) of PDLLA.

### 4.5. Droplet Microfluidics Processing

Particles were produced using a Telos^®^ High-Throughput System manufactured by Dolomite Microfluidics (Cambridge, UK). A single Telos^®^ module was used with a 2-reagent hydrophilic chip. The continuous phase was a 0.1% (*w*/*v*) PVA solution for experiments containing surfmer surfactants EGPEA-co-mPEGMA and THFuA-co-mPEGMA and a 2% (*w*/*v*) PVA solution for experiments producing particles with no polymer surfmer component. The dispersed phase contained a 5% (*w*/*v*) PDLLA solution in ethyl acetate with a 0.1% (*w*/*w*) surfmer component (THFuA-co-mPEGMA or EGPEA-co-mPEGMA) relative to ethyl acetate. For experiments not using any surfmer component, the dispersed phase only contained a 5% (*w*/*v*) solution of PDLLA in ethyl acetate. Emulsions were collected into DI water. Fluids were controlled used 2 Mitos P-Pumps connected to a control PC which were supplied by Dolomite Microfluidics. 250 µm FEP tubing with a 1.6 mm outer diameter (Dolomite Microfluidics, Royston, UK) was used to connected pressure pumps to the Telos module and also to connect the outlet to the collection vessel.

### 4.6. Particle Characterisation

SEM images of produced microparticles were taken at 15 kEV using a Hitachi TM3030 table-top scanning electron microscope (Hitachi, Krefeld, Germany). Images were taken at ×30, ×150, ×250, ×600 and ×1200 magnifications. Size analysis was performed on 3 regions of interest taken at ×250 magnification using ImageJ software.

A ToF-SIMS IV instrument (IONTOF GmbH, Münster, Germany) using a 25 keV Bi_3_^+^ primary ion source was used for surface chemistry analysis. Bi_3_^+^ primary ions were used with a target current of ~0.3 pA. Analysis for positive and negative spectra was acquired over a 500 µm × 500 µm scan area. Other analyses parameters were a cycle time of 100 µs, one shot/frame/pixel, one frame/patch and 20 scans per analysis. As the samples were of a non-conductive nature, charge compensation in the form of a low energy (20 eV) electron flood gun was applied. Images and spectra were acquired using SurfaceLab 6 software (IONTOF GmbH, Münster, Germany) and analysed using SurfaceLab 7.1 software (IONTOF GmbH, Münster, Germany).

## 5. Conclusions

In conclusion, this work has demonstrated the ability to stabilise biodegradable core particles with a bespoke surfactant chemistry. Micropipette characterisation identified the required concentration of PVA surfactant that was needed in addition to the functional surfactants to give longer term stability for particle generation. To minimise surface coverage of PVA, a 0.1% *w*/*v* PVA solution was found to stabilise the emulsion generation. Surface analysis was performed using ToF-SIMS and demonstrated the unique surfactant chemistry was visible at the surface with minimal PVA coverage. This optimisation process has shown how micropipette characterisation can minimise trial-and-error experiments by quantifying the interfacial tension across a concentration range. This methodology has enabled for the production particles with a unique surface chemistry which can in future be used to influence cell behaviour, whilst retaining desirable biodegradable bulk properties.

## Figures and Tables

**Figure 1 molecules-26-03302-f001:**
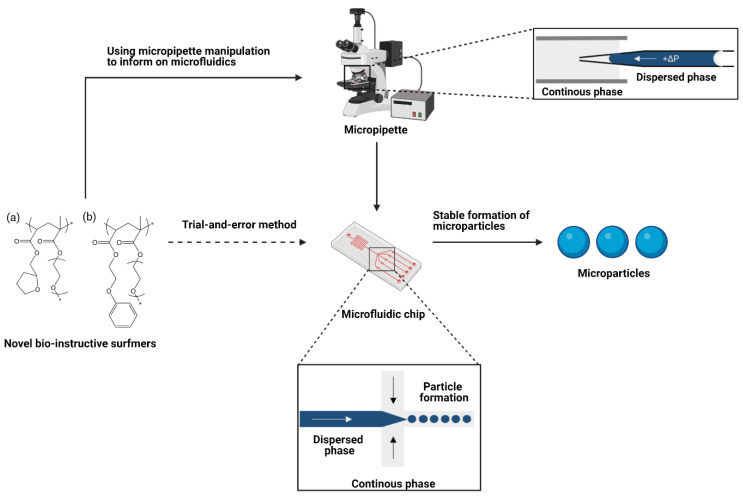
Particle formation with novel surfactants often requires lengthy optimisation if trial-and-error approaches are used. Using micropipette manipulation methods to inform droplet microfluidics removes this problem while providing detailed information about the system. The setup within the micropipette mirrors exactly that of the flow-focusing microfluidics chip so the information gained can be transferred directly. Polymeric surfactants used are (a) THFuA-co-mPEGMA and (b) EGPEA-co-mPEGMA in the dispersed phase with poly(d,l-lactic acid) (PDLLA) in ethyl acetate. The continuous phase is MilliQ water with, or without, poly(vinyl acetate-co-alcohol) (PVA). Figure created with BioRender.com.

**Figure 2 molecules-26-03302-f002:**
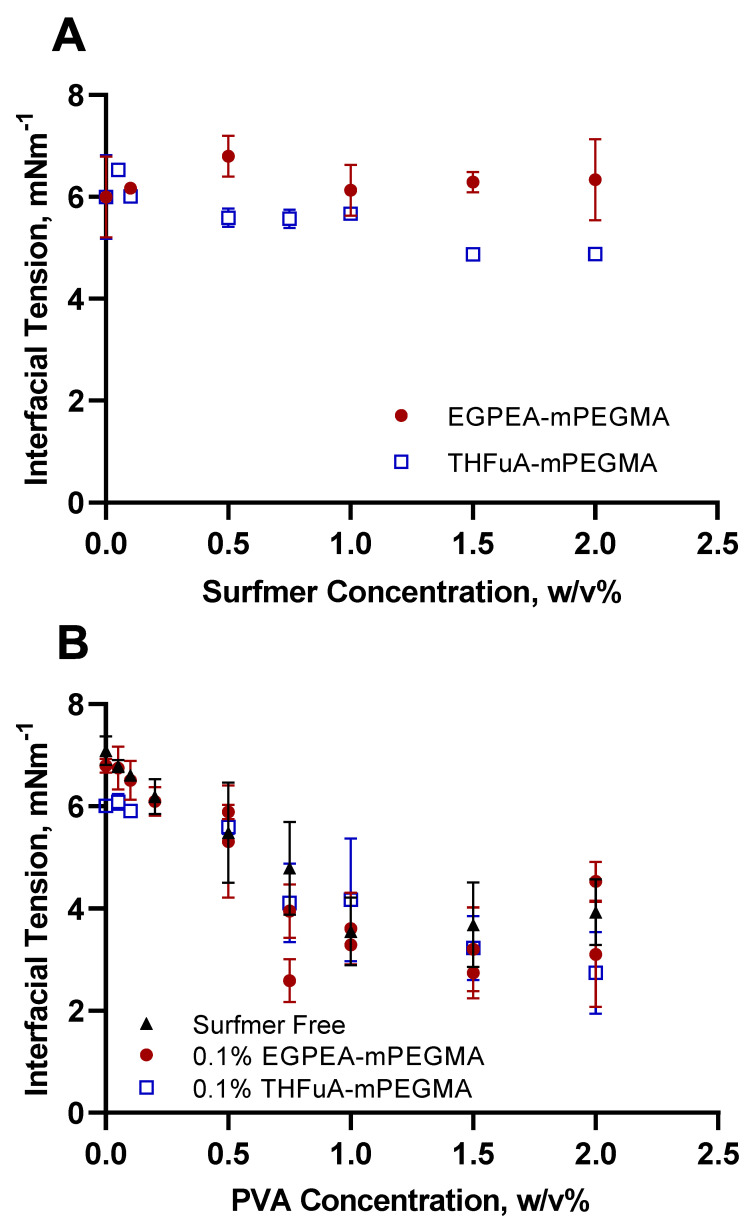
Interfacial tension of PDLLA (5% (*w*/*v*)) in ethyl acetate against water with the addition of (**A**) EGPEA-co-mPEGMA or THFuA-co-mPEGMA in increasing concentrations to the dispersed phase; (**B**) no surfmer or a fixed concentration of EGPEA-co-mPEGMA or THFuA-co-mPEGMA with increasing PVA concentration in the continuous phase. Error bars equal ± 1 SD unit, *n* = 3–4.

**Figure 3 molecules-26-03302-f003:**
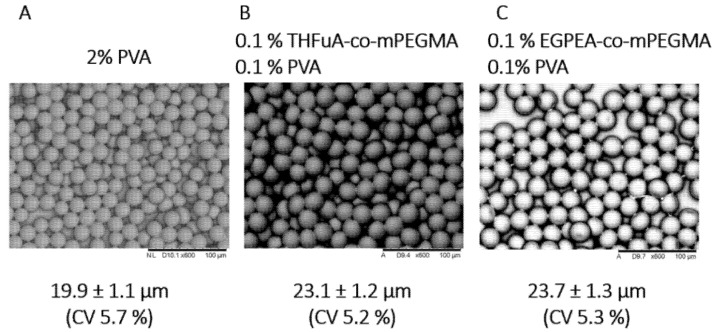
SEM images of particles produced using a droplet microfludics approach (**A**–**C**). Particles were manufactured with either PVA surfactant alone or with either of the bespoke surfmers THFuA-co-mPEGMA or EGPEA-co-mPEGMA with a weak 0.1% (*w*/*v*) PVA solution. Particles shown at 600× magnification.

**Figure 4 molecules-26-03302-f004:**
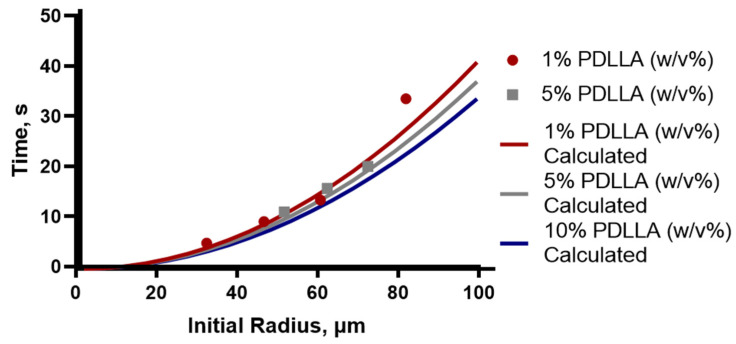
Solvent removal studies from the micropipettes were used to inform the collection vessel properties to reduce aggregation. The estimated time for solvent to be removed from drops of increasing size and concentration of PDLLA was calculated using a modified Epstein–Plesset model and shown by the red, grey and blue lines for 1%, 5% and 10% (*w*/*v*) PDLLA, respectively. The results of representative drops are shown in corresponding colours using circles and squares for 1% and 5% (*w*/*v*), respectively.

**Figure 5 molecules-26-03302-f005:**
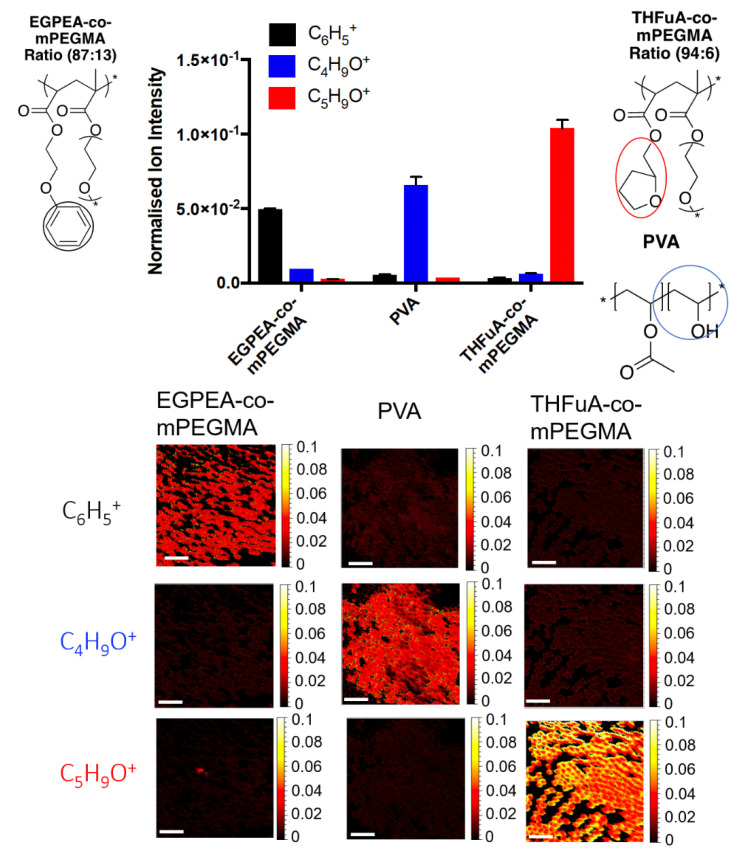
ToF-SIMS data showing intensities and chemical map images of three key ions associated to three surfactant structures (C_6_H_5_^+^–EGPEA-co-mPEGMA, C_4_H_9_O^+^–PVA and C_5_H_9_O^+^–THFuA-co-mPEGMA) where ions from the structures are circled in black, blue and red, respectively. Molar ratios of surfmers (hit material:mPEGMA) that have been included in the formulations were EGPEA-co-mPEGMA (87:13) and THFuA-co-mPEGMA (94:6). Particles containing surfmer were prepared using a dispersed phase containing 5% *w*/*v* PDLLA in ethyl acetate and a 0.1% *w*/*v* surfmer concentration. Emulsions were formed in a 0.1% *w*/*v* PVA concentration. Particles prepared without surfmer were prepared using a dispersed phase containing 5% *w*/*v* PDLLA in ethyl acetate. Emulsions were formed in a 2% *w*/*v* PVA concentration. Ion intensities have been normalised against total ion intensity. Error bars equal ± 1 SD unit, *n* = 3. Chemical map images are shown from a normalised intensity of 0 (black) to a normalised intensity of 0.1 (white). Scale bars on chemical images represent 100 µm.

## Data Availability

All relevant data are available from the University of Nottingham’s Research Data Management Repository.

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
