# Peer review of "Droplet Microfluidic Optimisation Using Micropipette Characterisation of Bio-Instructive Polymeric Surfactants"

_molecules, 2021, doi:10.3390/molecules26113302_

Round 1

Reviewer 1 Report

I think that the authors present a very interesting work wthat will be of interest to a wide audience of readers, and especially  to people working on microfluidic. The manuscript is very well written and schemes and figures very clear. In my opinion the manuscript should be publish as it is, with some minor corrections of the reference numbering in page 6.

Reviewer 2 Report

Review for Manuscript ID molecules-1224548

The research report focus on a very timely and relevant subject, the production of monodisperse biocompatible and (perspectively) biodegradable microparticles. The presented experimental results are credible, but the introduction to the subject and the interpretations and conclusions are not prepared appropriately. Overall, a substantial major revision is needed to make the manuscript readable, evident and conclusive.

The major issues are the following:

  1. The two surfmers, EGPEA-co-mPEGMA and THFuA-co-mPEGMA, are not adequately introduced in the introduction. Why were they chosen? Are they fully biodegradable? This seems unlikely, since PEG is claimed to be biocompatible, but not biodegradable. Currently, there is no ratio and logic in the design of the experiments, they “somehow fall from heaven” as do the conclusions – what will these systems serve for?
  2. The role of interfacial tension in the particle formation, monodispersity and stability against aggregation is not scientifically sound as is shown in several detailed comments below. Generally speaking, the interfacial tension between water (surface tension 72 mN/m) and PDLLA (surface tension about 5 mN/m in its molten form) is only modified by 0.1-1 mN/m in the range of concentrations used for the microparticles. It is very unlikely that these minor changes will have a substantial effect on particle size, monodispersity and aggregation stability. In case that the addition of the surfactants (surfmers) is effective with respect to the mentioned physical parameters, it has to be explored whether the underlying physical effect could be e.g. steric hindrance rather than the reduction in interfacial tension?
    In order to demonstrate the claimed effects clearly in the manuscript, it would be very helpful to show all parameters (size, size distribution, polydispersity, instability over time) in comparison also for pure PDLLA particles. In the current form, the claims are not yet acceptable as they do not contribute to the scientific elucidation of the underlying physical effects and their generalizability.
  3. The discussion about the relevance of the results is weak. What was the primary focus and goal of this study? How can the results be generalized and possibly applied for biomedical products or developments? Why is the discussion only focused on microparticles? PDLLA nanoparticles have much broader fields of applications.

The following detailed feedback is for specific aspects and paragraphs of the manuscript.

Introduction:

The following description is partially misleading:

“Most notably, microparticles are of significant interest in the fields of medicine and tissue engineering. Microparticles loaded with drugs, growth factors or proteins, are able to achieve specific and controlled delivery to cell cultures or as treatments, with the release of being tuned further by adjusting the size and internal structure of the particles [2–6].”

From the 5 articles cited, only ref. [2] refers to medical use while all others deal with in vitro or droplet engineering. Ref. [2] is a review from 2010 which states:

“As yet, there are very few microparticle drug delivery formulations approved for clinical use [64,65] although more clinical trials are currently underway [12,39]. The following articles present various microparticles designed for peptides and proteins delivery.”

The authors should:

  • Cite more up-to-date literature regarding the clinical relevance of microparticles
  • Be more clear about the clinical potentials and restrictions of microparticles

Fig. 1:

The terminology “surfmers (polymer surfactants)” first appears in the legend to this figure, but is not introduced here, which should be done in addition to the introduction in the text.

Results

Line 143-144: The sentence is unclear:

“The microfluidic strategy adopted was to create the dispersed phase by dissolving pre-synthesized PDLLA and one of two bespoke surfmers in ethyl acetate.”

What are the “two bespoke surfmers”? Please define. Reading further, it seems that EGPEA-co-mPEGMA and THFuA-co-mPEGMA are meant, but these were not introduced beforehand – please clearly state what you did and explain why.

Figure 2:

The title of x-axis is “Surfactant Concentration” while in the text the 2 polymers are termed “surfmers” – please unify. The same applies to line 249.

Lines 185-187:

“From Figure 2b, combining THFuA-co-mPEGMA with PVA, where PVA is kept at 0.1% (w/v) or below, provided a greater reduction in interfacial tension than either of the surfactants alone.”

The sentence is incomplete and it cannot be seen that it provided a greater reduction in interfacial tension as this effect does not seem to be statistically significant. The reduction of PVA to 0.1 w/v% should be quantified and compared to a regression of the two surfmer concentration curves by suitable statistical means (t-test or similar). In case significance can be proven, the sentence can remain and supported by such quantification. Otherwise, a convincing motivation for this parameter selection has to be provided as it currently seems to be a random choice.

Lost References

In lines 209 and 211 references are lost.

Figure 4 / Figure 5:

There are two figures termed “Figure 4”! The second is obviously Figure 5. This Figure 5 contains a lot of unclear information which has to be specified in the figure caption:

  1. what is meant by the ratios (87:13 and 94:6) of the two surfmers?
  2. To which standard have the “normalized ion intensities” (y-axis of the diagram on top) been normalized?
  3. At which surfmer or PVA concentration were the samples depicted in the bar diagram and the images prepared? What was the concentration of the other surfmer or PVA in the samples prepared with a specific agent – was it nominally zero? Are the quantified small amounts of these agents “background noise” or are they real contaminations?

Discussion:

This conclusion was supported by the observation that when similar surfmers were used in a solvent-free droplet microfluidics process, [44] concentrations approaching 2% (w/v) were needed to achieve stable emulsions.”

This comparison is much too short. Please define:

  1. what the “similar surfmers” are?
  2. what the similarities and differences of this experiment to the process shown in ref. [44] is?
  3. what concentration has to approach 2% - surfmers, PVA or PDLLA? What about the concentrations of the others? Currently, the discussion is very superficial and lacks precision.

Lines 265-267:

“The two surfmers show very different behaviour in terms of spread of the results: in most cases the standard deviation on the THFuA-co-mPEGMA measurements is an order of magnitude lower than EGPEA-co-mPEGMA, though both are within acceptable limits.”

What are “acceptable limits”? Please quantify and analyse statistically in order to draw a conclusion whether there are any statistically significant correlations between concentration and interfacial tension. In the case of the EGPEA-polymer, I don’t expect that any correlation can be shown which would contradict the cited statement.

Materials:

Line 327-328: The terminology is wrong and unclear:

"Throughout the experiments, poly(D,L-lactic acid) (PDLLA), Mn 47 000 gmol1, IV 0.5 dL g1 Evonik"

The exponents to physical dimensions have to be set as superscripts, not as subscripts. What is IV? Please define.

Reviewer 3 Report

Manuscript ID: molecules-1224548

Title: Droplet Microfluidic Optimisation Using Micropipette Characterisation of Bio-instructive Polymeric Surfactants

Comments:

In this paper, the authors adopted a micropipette manipulation method to optimize the concentration of bespoke polymeric surfactants to produce PDLLA microparticles, and investigated the effect of these surfactants on the interfacial tension and particle formation of the system. The topic of this work is interesting, presenting a promising approach in developing bio-microparticles based on droplet microfluidics. I can recommend the publication of this manuscript after revision. These are some main comments for this manuscript.

As shown in Figure 2A, the obtained IFT are based on the method of 4.3. Static Equilibrium Interfacial Tension Measurements. We can see that the IFT data is not significantly affected by the surfactant concentration, and the studied surfactant concentration is also very high, around 2%. So I am wondering whether the IFT shown in this figure is comparable to those measured by the common pendent drop and Wilhelmy Plate methods.

From Figure 2B, it can be seen that the presence of PVA leads to a significant reduction in interfacial tension, even at low concentration of 0.1%, and there is no significant impact on the IFT data after the addition of EGPEA-co-mPEGMA and THFuA-co-mPEGMA. Does that mean the PVA has a higher surface activity than EGPEA-co-mPEGMA or THFuA-co-mPEGMA? I am not sure whether the presence of these two bespoke surfactants is necessary for particle formation, or the PVA alone is sufficient to produce stable particles.

Line 246-248, actually, I did not get the point the authors would like to show. Please rewrite these sentences.

For the present used concentrations (up to 2%) of PVA or polymeric surfactants, the viscosity may be a factor influencing the accuracy of IFT measurements. The authors are suggested to give an explanation about this point.

Round 2

Reviewer 2 Report

Very careful revision, much appreciated!